# A Highly Homogeneous Airborne Fungal Community around a Copper Open Pit Mine Reveals the Poor Contribution Made by the Local Aerosolization of Particles

**DOI:** 10.3390/microorganisms12050934

**Published:** 2024-05-04

**Authors:** Sebastián Fuentes-Alburquenque, Victoria Olivencia Suez, Omayra Aguilera, Blanca Águila, Luis Rojas Araya, Dinka Mandakovic

**Affiliations:** 1Centro de Investigación en Recursos Naturales y Sustentabilidad, Escuela de Medicina Veterinaria, Facultad de Ciencias Médicas, Universidad Bernardo O’Higgins, Santiago 8370993, Chile; omayra.daniela86@gmail.com; 2Departamento de Matemáticas y Ciencias de la Ingeniería, Escuela de Ingeniaría Civil, Facultad de Ingeniería Ciencia y Tecnología, Universidad Bernardo O’Higgins, Santiago 8370993, Chile; 3Escuela de Biotecnología, Facultad de Ciencias, Ingeniería y Tecnología, Universidad Mayor, Huechuraba 8580745, Chile; victoria.olivencia@mayor.cl; 4Programa de Doctorado en Microbiología, Universidad de Chile, Ñuñoa 7800003, Chile; b.aguila.llanquilef@gmail.com; 5Fundación Ciencia y Vida, Huechuraba 8580704, Chile; 6Departamento de Química, Facultad de Ciencias, Universidad Católica del Norte, Antofagasta 1270709, Chile; l.rojas@ucn.cl; 7GEMA Genómica, Ecología y Medio Ambiente, Universidad Mayor, Huechuraba 8580745, Chile; dinka.mandakovic@umayor.cl

**Keywords:** particulate matter, airborne fungal community, fungi, bioaerosol

## Abstract

Fungi are ubiquitous and metabolically versatile. Their dispersion has important scientific, environmental, health, and economic implications. They can be dispersed through the air by the aerosolization of near surfaces or transported from distant sources. Here, we tested the contribution of local (scale of meters) versus regional (kilometers) sources by analyzing an airborne fungal community by ITS sequencing around a copper mine in the North of Chile. The mine was the regional source, whereas the soil and vegetal detritus were the local sources at each point. The airborne community was highly homogeneous at ca. 2000 km^2^, impeding the detection of regional or local contributions. Ascomycota was the dominant phylum in the three communities. Soil and vegetal detritus communities had lower alpha diversity, but some taxa had abundance patterns related to the distance from the mine and altitude. On the contrary, the air was compositionally even and unrelated to environmental or spatial factors, except for altitude. The presence of plant pathogens in the air suggests that other distant sources contribute to this region’s airborne fungal community and reinforces the complexity of tracking the sources of air microbial communities in a real world where several natural and human activities coexist.

## 1. Introduction

Aerosols are colloidal systems of solid or liquid particles suspended in a gas [1]. Aerosols are emitted into the atmosphere from natural (e.g., erosion of surfaces, soil dust, sea breeze) or anthropogenic sources (e.g., particulate matter near urban, industry, or mining centers). These emissions are particularly high in human-impacted landscapes due to big urban, industrial, or mining activities [2,3,4]. Microorganisms are present in the atmosphere in the form of bioaerosols, i.e., particles of biological nature with diameters ranging between 1 nm and 100 μm, which include living and dead organisms such as bacteria, archaea, and algae, as well as dispersal units such as fungal spores or plant pollen; and larger biological fragments such as plant detritus or insect excretions. Bioaerosols can account for 20–80% of aerosol particles [2] and include a myriad of microorganisms dispersed by air.

Airborne microbial communities follow seasonal, altitudinal, temporal, and spatial patterns like other environments [5,6,7,8,9]. For instance, spatial similarity distance-decay of air microbial communities has been observed for bacteria and fungi at scales of just 500 m [10,11]. However, other authors could not find such a relationship at larger scales, neither in air nor soil [12], which implies a high dispersal [13,14]. Global scale studies suggest that spatial distribution depends on seasonality, meteorological factors, close environments, emission sources, etc. [15,16,17,18]. Fungal and bacterial diversity in urban areas is lower than in rural zones [19], with higher total cultivable fungi on days with lower particulate matter (PM) pollution [20]. However, some genera, such as *Alternaria* and *Epicoccum*, can increase their abundance during high (PM) days [21]. The movement of microorganisms through the atmosphere is far to be determined and remains a major challenge. Regarding public health and environmental risk assessment, tracking the load of microorganisms we breathe still lacks information.

Fungi is a ubiquitous and ecologically versatile kingdom that can be dispersed through the air as hyphae fragments, vegetative yeasts, or spores [14]. Fungi dispersed by air include allergens, plant pathogens, saprophytes, or opportunistic animal/human pathogens [14,22]. Some fungal genera are particularly resistant to extreme environmental stressors such as heavy metals, cold, UV radiation, or osmotic stress [23,24,25]. The airborne fungal community (FC) in a determined point is influenced by the aerosolization of near surfaces or the bioaerosols transported/deposited from other distant sources. In addition, factors such as wind speed and direction, temperature, barometric pressure, UV radiation, and precipitation influence fungal dispersion. This leads to the obvious question: at any time and space, how much airborne fungi is coming from the uplifted particles from the near-surface, and how much from other faraway places? Which environmental and spatial drivers are influencing this community? These questions have important scientific, environmental, health, and economic implications.

This study used a sampling strategy around a copper mine—a well-known source of aerosols by the generation of PM—as a model to address this problem. Near mine tailings or smelters, gradients of metal concentration in soils and aerosols can arise from the source. For instance, the soil’s copper, zinc, lead, cadmium, and mercury concentrations can decay 1 to 2 orders of magnitude less than 30 km from the source [26,27]. Similarly, metal concentration in the air appears to decay in less than 3 km and is highly dependent on wind conditions [28]. The North of Chile has harbored intense mining activity for centuries. Studies usually focus on health issues related to mining activities, such as the high levels of PM in the indoor air in schools due to aerosols emitted from mine tailing deposits [3]. Some surveys have tracked the distribution, for instance, of mercury in the air [29] and copper, zinc, and arsenic in stream sediments and soils in the Coquimbo Region [30,31]. These metals are present naturally or through mining activity in the North of Chile. However, the distribution of microorganisms associated with PM is absent in the literature.

Andacollo is a town of ca. 9500 inhabitants in the North of Chile (30.26146° S, 71.08166° W, 1070 m.a.s.l.). The area has been known for its copper and gold deposits since the XVII century. The town is immediately next to a copper and gold open pit mine that has been intensively exploited since the 1890s. The Chilean Government declared the area “saturated” of PM10 in 2009, and a decontamination plan started in 2015. Because of this well-known static PM10 source, it is considered a suitable model for testing the influence of a PM10 source on airborne FC and its dispersion. Here, no influence of the mine was found, suggesting that other forces are shaping fungal communities in this environment.

## 2. Materials and Methods

### 2.1. Study Design

The sampling area consisted of a central element (i.e., the mine), and sampling points were distributed concentrically around this element (Figure 1). Three sample types were collected at each point, including air, soil, and dry vegetal detritus, to assess the local contribution (i.e., the scale of meters) of airborne FC due to the aerosolization of either soil or vegetal detritus at each point. The regional contribution (i.e., scale of kilometers) was assessed by the resemblance of air communities with respect to the mine at different distances. At each point, a portable GPS device recorded geographical coordinates and altitude (Garmin GPSMAP 64s). This allowed us to calculate the longitudinal distances (km) between pairs of samples and the differences in altitude (m.a.s.l.). At each sampling point, temperature, wind speed and direction, humidity, and UV radiation were recorded with a meteorological station (Vantage Pro Plus, Davis Instruments). Two sampling campaigns were performed in the spring (September–October 2021) and summer (January 2022). In each season, 35 sites totaled 70 air, 70 soil, and 61 vegetal detritus samples (*n* = 201). Full details are provided in Appendix A.

### 2.2. Sampling of Air, Soil, and Vegetal Detritus

Each site consisted of an app. 10 × 10 m square. In the square center, aerosols were collected with a Coriolis μ sampler (Bertin Technologies, Montigny-le-Bretonneux, France) placed 1.5 m from the soil. The air was sampled at a flow rate of 300 L min^−1^ for 2 h in 15 mL of sterile bi-distilled water of clinical grade (Fresenius Kabi Apiroflex, Sanderson Inc., Santiago, Chile) as a collector liquid. After collection, the suspension of microorganisms and particles was filtered through a sterile 0.22 μm polyether sulfone filter (Whatman Puradisc 25, Merck, Buckinghamshire, UK) with a sterile syringe. The soil samples consisted of a composite topsoil (0–5 cm depth) sample composed of 5 randomly collected subsamples (app. 1–2 m distant distributed in the square). Similarly, the composite vegetal detritus sample comprised 5 randomly collected subsamples (app. 1–2 m distant distributed in the square). In each case, the composite sample was mixed in a sterile 50 mL polypropylene tube. All samples were stored at −20 °C during the sampling campaign and were further stored at −80 °C in the laboratory until DNA extraction.

### 2.3. Determination of Metals in Soil

Aluminum (Al), cadmium (Cd), copper (Cu), iron (Fe), lead (Pb), and zinc (Zn) were determined by acid extraction assisted by a microwave (Ethos Easy, Milestone Connect). The same treatment was performed with ERA-540, metals in the soil as Certificated Reference Material (CRM, Environmental Resource Associates, Golden, CO, USA). A dried mass of 0.5 g of soil and CRM were weighted in the microwave vessels and digested using 10 mL of HNO_3_ 65% (Merck Cat.No. 1.00456), 5 mL of HClO_4_ 72% (Merck Cat.No. 1.00519), 3 mL of HCl 37% (Merck Cat.No. 1.00317) and 2 mL of HF 40% (Merck Cat.No. 1.00338, Darmstadt, Germany). The microwave was set at 200 °C for 60 min and then cooled until room temperature. The resulting digestions were filtered using a CN-542 quantitative paper filter (Microclar, Buenos Aires, Argentina) and diluted until 100 mL in volumetric flasks. Blank controls were also included. Soils, CMRs, and Blanks were digested, and several dilutions were applied to fit into a multi-elemental calibration curve from 1 to 100 ppb. The determination of the content of metals in the diluted solutions was performed by Inductively Coupled Plasma–Optic Emission Spectrometry (ICP-OES 5900, Agilent, Santa Clara, CA, USA) in triplicate with the following parameters: read time 5 s; sample uptake delay 25 s; stabilization time 15 s; rinse time 60 s; pump speed 12 rpm; RF power 1.2 kW; auxiliary flow 1 L/min; plasma flow 12 L/min; nebulizer flow 0.7 L/min; viewing height 8 mm; plasma viewing axial/radial system; and the nebulizer-type sea spray/double pass cyclonic spray chamber. Mercury was determined directly in the soil using a Mercury Analyzed DMA-80 evo (Milestone Connect, Bergamo, Italy).

### 2.4. DNA Extraction, ITS1 Amplification and Sequencing

Total community DNA was prepared using the PowerSoil DNA Isolation Kit (QIAGEN, Hilden, Germany). For soil and vegetal detritus samples, 0.5 g was used as starting material, and the manufacturer’s instructions were followed. Air samples were handled in a laminar flow cabinet to avoid the risk of contamination due to the low biomass. Each filter was cut with sterile surgical scissors into small pieces and added to the kit’s glass beads tube, and the manufacturer’s protocol was followed. The final 100 μL volume was concentrated by lyophilizing to 20 μL. In all cases, a microtube homogenizer (BeadBug 6, Benchmark, Suzhou, China) aided the mechanical disruption step by 4 pulses of 90 sec at 4350 m/s. Purified DNA was used as a template to amplify the fungal internal transcribed spacer 1 (ITS1), which was amplified and sequenced according to the Earth Microbiome Project protocols (www.earthmicrobiome.org/protocols-and-standards, accessed on 10 April 2024) [32]. The PCR amplification was performed using the primers ITS1f (CTTGGTCATTTAGAGGAAGTAA) and ITS2 (GCTGCGTTCTTCATCGATGC), which yields a PCR product of 250 to 600 bp [33] plus the Illumina adapters. Amplification products were multiplexed and sequenced in the Illumina MiSeq platform (2x250bp) at Argonne National Laboratory (Lemont, IL, USA) as described previously [34]. The sequences were submitted to the NCBI BioProject database with the project identification number PRJNA1097653 and sample accession numbers SAMN40875613 to SAMN40875682 (air), SAMN40903137 to SAMN40903206 (soil), and SAMN40904544 to SAMN40904604 (vegetal detritus).

### 2.5. Processing of Raw Sequences and Data Analysis

The processing from raw sequences to contingency tables was performed with QIIME2 v2020.2 [35]. Each dataset was demultiplexed into per-sample sequences, which were quality-checked, assembled, and chimera-filtered with DADA2 [36] using default parameters. With the DADA2 strategy, we obtained unique amplicon sequence variants (ASV). Taxonomy was assigned using the UNITE fungal database v9.0 [37] with vsearch [38]. The reads in the dataset classified as plants and non-fungi were discarded. The resulting ASV table consisted of 12,267 ASVs in 194 samples; 7 samples were discarded because of the low number of reads. The ASV table was rarefied to 2421 reads per sample to calculate alpha- and beta-diversity metrics. For weighted Faith’s phylogenetic diversity (PD) and UniFrac calculation [39,40], sequences were first aligned with MAFFT [41], and the tree was constructed using Fasttree [42]. All steps were performed with the software versions implemented in QIIME2. Correlation analyses of environmental drivers and diversity metrics were done using Spearman correlation; Mantel tests were used for beta-diversity. Rank abundance and similarity decay plots were performed with BiodiversityR v2.15-4, geosphere v1.5-18, and betapart v1.6 packages in R v4.3.2. Procrustes analysis [43] was performed in QIIME2 software.

## 3. Results

A total of 12,267 ASVs across 194 samples were obtained. Seven samples had low sequencing depth (<2421 reads) and were discarded from the analyses. Richness was significantly higher (Kruskal–Wallis *p* < 0.001) for air communities, with a median of 135 ASVs, which contrasted with the 60 and 70 ASVs found in soil and vegetal detritus communities, respectively (Figure 2a). Faith’s phylogenetic diversity and the Shannon index yielded the same result (Appendix A). Thus, independent of the nature of the metric, the result pointed in the same direction: the diversity of airborne FC was significantly higher than soil and vegetal detritus FC. The ASVs in the air community were evenly abundant, with the most abundant ASV in ca. 2% of the community; for soil and vegetal FC, the most abundant ASV dominated the community in ca. 7% and 14%, respectively (Figure 2b).

Beta diversity analysis shows that the three communities significantly clustered as different groups (Figure 3a, PERMANOVA *p*-value < 0.001). Airborne FCs were highly homogeneous, i.e., with high similarity among samples. The points corresponding to airborne FC formed a tight cluster with lower dispersion (PERMDIST *p*-value < 0.001) than soil and vegetal detritus FC. This was more evident for the Jaccard (Figure 3a) and Bray–Curtis tests (Appendix A), which accounted for shared members in the community. Although significant, the differences in dispersion were not marked as such in the PCoA plots for phylogenetic metrics (Appendix A). Considering the homogeneity of airborne FC, we further tested if beta diversity was enough to observe spatial patterns. Airborne FC showed no similarity to distance–decay with all the metrics tested. On the other hand, soil and vegetal detritus had a classical decaying relationship. This pattern was stronger for Jaccard (Figure 3b) and Bray–Curtis (Appendix A), whereas phylogenetic metrics had weaker slopes (Appendix A).

Due to the absence of a spatial pattern of airborne FC, we tested the influence of local sources, i.e., the aerosolizing particles of soil or vegetal debris, at each sampling point. Using Procrustes analysis, we tested if airborne and soil FCs were related. The test compared the shape of the two groups of samples by rotating and resizing one to make the shapes fit. The Procrustes results yielded significant (*p* < 0.001) but weak goodness-of-fit values for the air–soil (M^2^ = 0.840), air–vegetal (M^2^ = 0.836), and soil–vegetal (M^2^ = 0.629) datasets (Figure 3c–e). We further tested the correlation of the beta diversity distance matrices using the Mantel test; no statistically significant correlation was found between each pair of communities (not shown).

To identify the drivers of airborne FCs, relevant environmental variables such as air humidity, temperature, pressure, solar radiation, and UV dose were correlated (Spearman) with beta diversity. During the sampling periods, the recorded humidity was 45.4 ± 16.1%, temperature was 21.2 ± 5.2 °C, solar radiation was 783 ± 252 W m^−2^, and pressure was 723 ± 25 mmHg. The spatial variables altitude and distance to the mine were also included in the analysis. The altitudes of sites ranged from 328 to 1341 m.a.s.l., and the distance to the mine ranged from 0 (sites at the mine) to 27 km (the farther sites). In addition, because of the mining activity, metals were quantified in soils and correlated with the soil FC. The concentrations were in the following ranges: aluminum 6904 ± 4175 mg kg^−1^, copper 213 ± 144 mg kg^−1^, iron 75,843 ± 18,117 mg kg^−1^, mercury 292 ± 235 mg kg^−1^, lead 43 ± 56 mg kg^−1^, and zinc 119 ± 80 mg kg^−1^. Cadmium levels were below the detection limit in all sites. The full record of environmental and spatial drivers is shown in Appendix A.

The results are shown in Table 1, where variables with at least one significant correlation (*p*-value < 0.001) are included. Weak correlations were found for altitude, distance to the mine, and pressure. Altitude yielded the highest correlation values for vegetal detritus FC (ρ = 0.4523 and 0.3976 for Jaccard and unweighted UniFrac, respectively). This relationship was weaker for the soil (ρ = 0.2790 and 0.1835 for Jaccard and unweighted UniFrac) and air (ρ = 0.1468 and 0.1367) communities. Only the vegetal detritus FC was correlated with the distance to the mine (ρ ~ 3 with all metrics). Finally, barometric pressure was correlated with air FC with similar values to altitude. Despite the mining activity in the area, no significant correlations were found between any metal and soil FCs. Detailed correlation results are presented in Appendix A.

Considering that the only spatial variables were significantly correlated with beta diversity (Table 1), taxonomic composition is summarized in the samples grouped according to altitude and distance to the mine (Figure 4). Ascomycota largely dominated the fungal communities, with Basidiomycota and an unidentified phylum representing ca. 10 to 20% of the community members. Basidiomycota was represented by *Naganishia* (Filobasidiaceae) in ca. 8% of the total community. A Capnoidales genus (light blue) was the dominant taxa in the air and vegetal detritus communities. *Alternaria*, *Naganishia*, Dothideomycetes, Teratosphaeriaceae, Helotiales, Pleosporales, Dothioraceae, and *Cladosporium* followed in relative abundance. Together, these nine taxa represented ca. 75% of air and vegetal detritus FC and ca. 50% of the soil community. The composition was consistent with PCoA and similarity–decay plots: no evident pattern can be observed for airborne FC, supporting the homogeneity of this community. On the contrary, some gradients appeared for soil and vegetal detritus communities in relation to altitude and the distance to the mine. Helotiales (red) in soil and Dothioraceae (strong orange) in vegetal detritus had a similar distribution: they had a higher proportion at higher altitudes and were closer to the mine. *Alternaria* (blue) in soil and the Capnoidales genus (light blue) in vegetal detritus were in the opposite pattern, with higher proportions at lower altitudes farther from the mine (Figure 4). Finally, Teratosphaericeae and *Cladosporium* were at a higher proportion in the airborne FC than soil and vegetal detritus communities. These two taxa belong to the Capnoidales order and, together with the Capnoidales unassigned genus (light blue), represented ca. 40% of the air community.

## 4. Discussion

### 4.1. Airborne Fungal Communities Are Diverse but Homogeneous at the Regional Scale

In the present study, we assessed the airborne FC in Andacollo. The town is next to a copper and gold open-pit mine in the North of Chile. Because of this static PM10 source, we studied it as a model for testing its influence on the airborne FC. In addition to air, soil and vegetal detritus samples were taken at each sampling point in a ca. 25-km radius area because they can contribute to the airborne FC by the aerosolization of close surfaces. By this sampling strategy, we tested the influence of sources at a local scale (i.e., meters) versus the regional scale (i.e., kilometers) on the airborne community (Figure 1).

The first exploratory approach was the alpha-diversity of communities (Figure 2). The higher diversity of airborne FC was unexpected because the soil is a particularly diverse environment, whereas air is classically considered only a pass-through environment where conditions are unfavorable for life. However, airborne FC is apparently as diverse as those in other environments. Some authors have found that airborne FC is more diverse than soil in rural and urban landscapes [44], whereas others have found less in the air than in the soil [45]. A common pattern is that human activities related to urbanization reduce fungal diversity and increase the abundance of Ascomycota [44,46], the dominant phylum worldwide [47]. Although Andacollo is a small town, decades of mining have greatly impacted the entire area. Therefore, Ascomycota dominance (Figure 4) was somehow expected due to the human activities taking place. However, air, soil, and vegetal debris FC differences arise when the composition is analyzed at lower taxonomic ranks.

The most abundant genus in the air was *Alternaria*, which includes some plant pathogens and mycotoxin-producing species and is relevant for public health because of its allergenicity [48,49,50]. This genus and the Capnoidales unassigned genus had the same pattern in soil and vegetal detritus composition related to altitude and distance to the mine (Figure 4). Capnoidales was clearly enriched in the air community, with Teratosphaericeae, *Cladosporium*, and the unassigned genus as the most abundant taxa. Teratosphaericeae includes 61 genera with various features, including plant pathogens and saprophyte species, extremophiles, human opportunistic pathogens, and lichen-associated [51]. Thus, without deeper taxonomic resolution, assessing which ecological feature could be related to this group is impossible. *Cladosporium* spp. are ubiquitous filamentous fungi typically non-pathogenic to humans, although it has been reported to cause infections in immune-compromised patients [22]. *Cladosporium* spp. have been isolated from metal-polluted soils [23], and, together with *Alternaria*, spores of this common mold are found in the air and regarded as allergens [14]. Thus, probably much of the *Cladosporium* and *Alternaria* in the air were actually spores. The genus *Naganishia* (formerly *Cryptococcus*) was evenly distributed in the three communities, so it is apparently ubiquitous and not dependent on a specific environment. This genus includes extremophilic yeasts studied as model organisms. Members of this genus encode proteins associated with psychrotolerance, osmotolerance, UV, and dehydration/desiccation resistance [24]. *Naganishia* spp. have been described in the north of Chile, so its presence in the sampled area is expected [25]. Finally, *Botrytis cinerea*, a well-known grapevine pathogen, was present in ca. 2% of air but absent in soil and vegetal debris communities, showing plant pathogens’ possible dissemination and impact through air. The study area is between two agricultural valleys, so other influences not tested in this work can also influence the air community. The presence of *Alternaria* and *Botrytis* in the air opens new aspects about the fungal dispersion and their consequences that need further research.

Finally, the alpha-diversity based on ASV showed a completely different aspect of the communities than the one observed regarding taxonomic composition. The air community had higher alpha-diversity, and ASVs were evenly distributed, i.e., none dominated the community (Figure 2 and Appendix A). However, most ASVs were affiliated with few taxa, i.e., nine taxa accounted for 75% of the air community (Figure 4).

### 4.2. Tracking the Drivers and Sources of Airborne Fungal Communities

Airborne FC follows seasonal patterns and is dependent on the sampling site, i.e., they are defined by local contributions [52]. Here, the airborne fungal community was not related to seasonal variations or other categorical factors (not shown) reported consistently in the literature [6,13,50,52]. Airborne FC diversity was not correlated with classical environmental drivers such as air temperature or humidity. The general rule is that water availability allows the development of a major number of species, and temperature is a major environmental factor that modulates growth. Mean rain precipitation and humidity are normally found to modulate FC in the air [53], whereas temperature drives fungal distribution worldwide [54]. A wider range is probably necessary to observe such relationships in this region. Notably, the well-known stressors for microorganisms and, generally, for life—solar/UV radiation in air and metals in soil—had no relationship with fungal alpha- or beta-diversity. Solar radiation was in the range typically registered in this area in these seasons by the National System of Information on Air Quality [55]. Metal concentrations were in ranges similar to the previously reported [30,31]. Mercury, which has been used in old-fashioned gold recovery techniques, is particularly elevated in this area, as reported by the present study and previous surveys [29]. It has probably been accumulated historically by deposition of PM or stream sediments, a mechanism of dispersion for metals and inorganic contaminants [18,28]. Although heavy metal contamination is a major stressor that may be contributing to microbial community diversity and distribution, this relationship could not be established in this study. The only significant environmental driver for airborne FC was barometric pressure, which is obviously related to altitude. Actually, Spearman’s correlation values are similar for both drivers (Table 1). The altitude was the only variable correlated consistently with all communities and metrics. Correlation with vegetal detritus beta-diversity yielded the highest values, consistent with the taxa distribution at different distances from the mine shown in Figure 4. Overall, the results suggest that spatial factors, instead of environmental factors, drive the observed differences in soil and vegetal detritus FC composition (Table 1).

Mantel tests between distance to the mine and airborne FC yielded no significant correlations, which agrees with the flat line in the similarity decay plot (Figure 3b). Similarity decay describes a negative relationship between community similarity and longitudinal distance, widely reported in multiple environments, including soils, water bodies, and air [56]. Here, only soil and vegetal detritus had the expected negative slope. Similarity decay depends on the ecological context, but a recent meta-analysis concluded that microbial ecological patterns are distorted by methodological choices [56]. This is true due to the coexisting number of techniques to study microbial communities. Each has its own bias, such as culture conditions and media (culture-dependent methods) or studied gene, primer selection and sequencing depth (culture-independent methods), etcetera.

Nonetheless, in this study, the methods were standardized before sampling. DNA was extracted using the same method for all samples (a validated commercial kit), and the PCR and sequencing were performed at the same facility. All these deliberated cares to avoid methodological biases suggest that differences are attributed to ecological context. Apparently, the high degree of dispersion tends to homogenize communities, resulting in a higher proportion of shared members among samples. This phenomenon has been observed at regional scales in California, spanning 40 km [12] and 75 km [57] distances. Like the present work, Wagner et al. (2022) found no relationship between airborne and soil FC and described a highly homogeneous airborne FC with no similarity decay [57]. At the global scale, however, the surrounding landscapes control the airborne FC [15]. Together, these surveys suggest that fungi in the air follow patterns only at higher scales.

The high degree of homogeneity of airborne FC impeded us from tracing the source of the aerial fungi in this study. So, to check the local contribution, we used Procurustes analysis to visually test the fit of the points in a PCoA ordination. The test is commonly used to fit two datasets, for instance, from two sequencing platforms or analyzed by two methods [58,59]. The idea is to test how good the overlapping of the PCoA points is by fitting the shape of the group of points. Procrustes can assess the match for individual observations (e.g., two samples from the same site in our case), which is not available with the widely used Mantel test [43]. Although there is no absolute cutoff value for the goodness-of-fit M^2^, the statistic can take values between 0 and 1. Lower M^2^ values indicate a better fit [60], and a good fit is usually considered when M^2^ < 0.3 [43,58,59,60]. We were far from this value; thus, the relationship between airborne FC and soil/vegetal FCs was weak. This discarded a meaningful contribution of local sources. However, the regional contribution could not be determined either. Other authors have shown that the influence of local sources can be as high as 60%, with less than 4% of species coming from distant sources [5]. However, the observed homogeneity of airborne fungal communities is consistent with other studies where the sampling site is not relevant [12,61], even at distances as large as 900 km [13]. A recent study assessed the influence of the Saharan–Sahelian dust event on respirable metals and microorganisms in Texas, USA, i.e., on a global scale. Although elemental composition could be tracked from African sources, the airborne bacterial and fungal communities correlate with local sources such as vehicular emissions, construction activities, and, notably, calcium content [18]. Here, the influence of the mine on airborne fungal communities could not be established. Andacollo has been under a PM10 decontamination plan for eight years, which consists of the wetting of surfaces to avoid the generation of PM10, particularly during blasting and high truck traffic. Therefore, it is expected that the PM10, at least in terms of fungal distribution, is not influenced by the mine. Instead, a highly homogeneous, diverse, and compositionally even airborne fungal community was observed. The airborne community neither followed spatial similarity decay nor correlated with environmental variables in a ca. 25 km radius study area, suggesting that other forces are shaping fungal communities in this environment and/or the number of samples and/or seasons we analyzed were not enough to identify such patterns. This reinforces the complexity of tracking the sources of air microbial communities in the real world, where several natural and human activities coexist.

## 5. Conclusions

Classical environmental drivers were found to not influence airborne fungal distribution in the studied area. The airborne fungal community was not related to either local or regional contributions. Instead, a highly homogeneous and compositionally even community was observed, where altitude was apparently a significant diversity driver. The presence of plant-associated genera suggests that other faraway sources contribute to the airborne fungal community in this area.

## Figures and Tables

**Figure 1 microorganisms-12-00934-f001:**
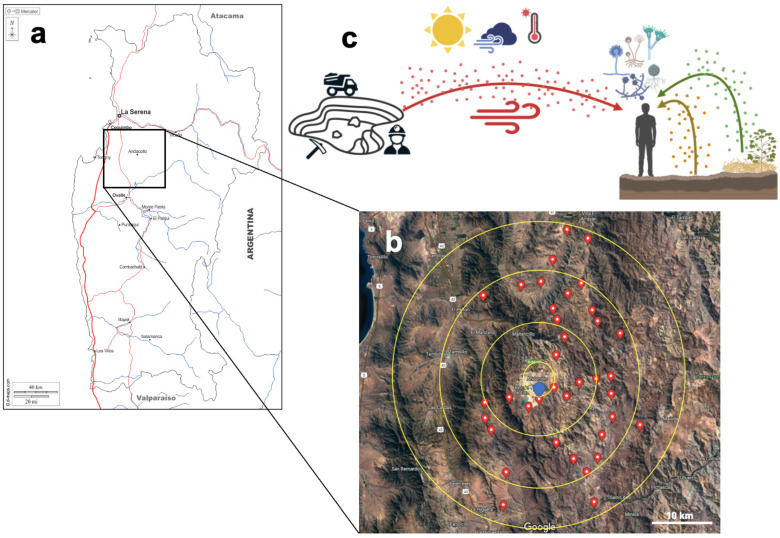
Sampling area at the Coquimbo Region, North Chile (**a**). The town of Andacollo and its mine (30.2614 °S 71.0816 °O) are located at the central point depicted in blue (**b**). Starting from this central point, several points were sampled to encompass an approximately 25 km radius. The upper panel shows the sampling strategy used in this study (**c**). Three sample types were collected at each point: air (red), topsoil (brown), and vegetal detritus (green). Environmental variables and geographical coordinates were recorded at each point.

**Figure 2 microorganisms-12-00934-f002:**
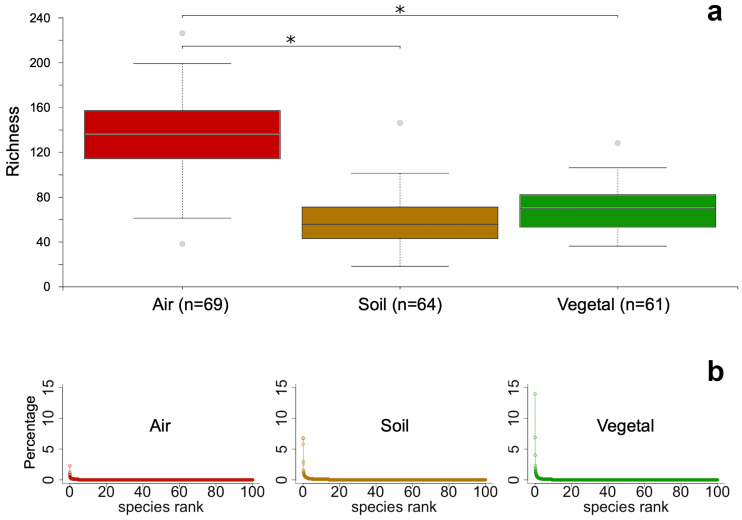
Alpha diversity of the three communities: air (red), soil (brown), and vegetal detritus (green). (**a**) Richness was calculated as the number of ASVs in each community; the median is depicted in the center of each box, and the outliers are shown as gray circles. Asterisks show significant differences (*p* < 0.001) through the Kruskal–Wallis pairwise test. (**b**) Rank-abundant distribution of ASVs in each community.

**Figure 3 microorganisms-12-00934-f003:**
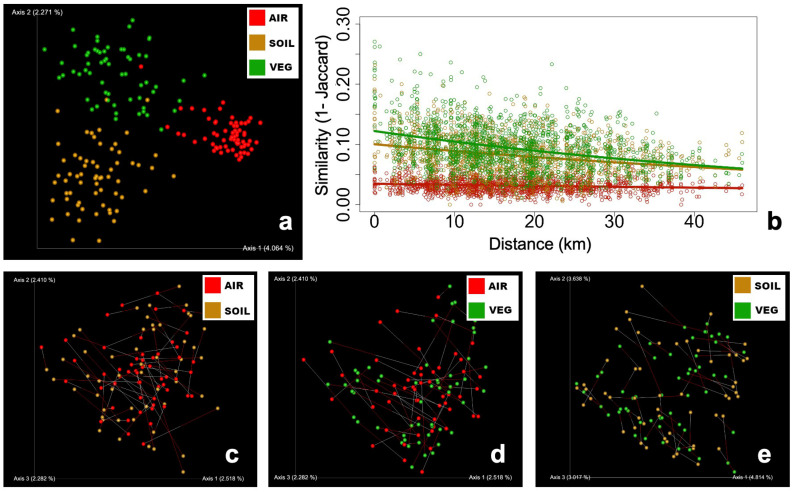
Beta diversity (Jaccard distance) analysis of the three communities: air (red), soil (brown), and vegetal detritus (green). (**a**) Principal coordinates plots, where the three communities are significantly grouped in different clusters. (**b**) Similarity decay plot shows that airborne communities had no relationship with respect to geographical distance. (**c**–**e**) Procrustes analysis of the air–soil, air–vegetal, and soil–vegetal communities.

**Figure 4 microorganisms-12-00934-f004:**
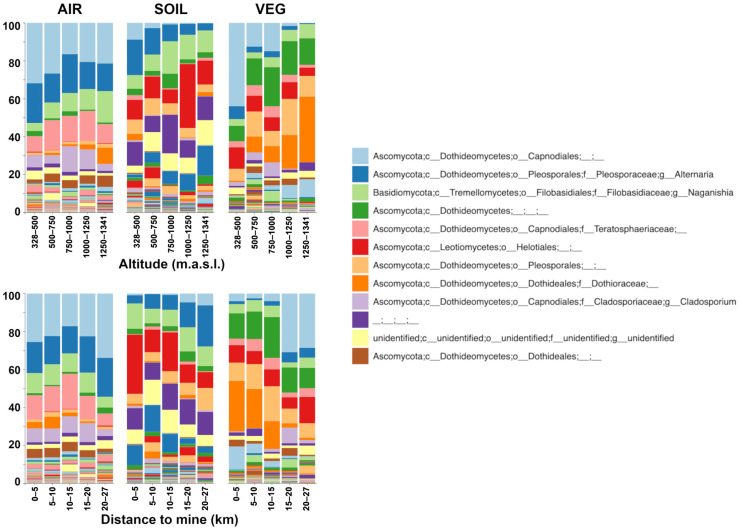
Taxonomic composition (genus level) of the air, soil, and vegetal detritus communities. Samples were grouped by altitude (upper panel) and distance to the mine (lower panel). The legend shows only the 12 most abundant taxa.

**Table 1 microorganisms-12-00934-t001:** Mantel’s test of beta diversity and environmental/spatial variables. Values are Spearman’s correlation coefficient (ρ). Only *p* < 0.001 significant correlations in at least one sample type are shown. n.s., not significant.

Variable	Beta Distance Metric	Air	Soil	Veg
Altitude (m.a.s.l.)	Jaccard	0.1468	0.2790	0.4523
	unweighted UniFrac	0.1367	0.1835	0.3976
Distance to mine (km)	Jaccard	n.s.	n.s.	0.3240
	unweighted UniFrac	n.s.	n.s.	0.2838
Pressure (mmHg)	Jaccard	0.1527	n.s.	n.s.
	unweighted UniFrac	0.1302	n.s.	n.s.

## Data Availability

The sequences were submitted to the NCBI BioProject database with the project identification number PRJNA1097653 and sample accession numbers SAMN40875613 to SAMN40875682 (air), SAMN40903137 to SAMN40903206 (soil), and SAMN40904544 to SAMN40904604 (vegetal detritus).

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
