# Peer review of "A Highly Homogeneous Airborne Fungal Community around a Copper Open Pit Mine Reveals the Poor Contribution Made by the Local Aerosolization of Particles"

_microorganisms, 2024, doi:10.3390/microorganisms12050934_

Round 1

Reviewer 1 Report

Comments and Suggestions for Authors

The work presented in this article is extensively documented, so the authors provide sufficient information in the introduction to state the objective clearly and unambiguously.

The analytical methodology established includes the most used techniques for the determination of metals, so the results obtained are in accordance with the discussion raised and, therefore, with the conclusion of this work.

The results are presented using figures and graphs that are clear to reflect the correlations indicated by the authors.

No plagiarism was found within the writing and no self-citations were used. All references are appropriate for the research work.

I consider that the work can be published without major revisions.

Reviewer 2 Report

Comments and Suggestions for Authors

The article studied the influence of copper open-pit mine and the dust (Particle matter) they generate on the diversity of fungal communities in the air, soil, and vegetal detritus within a radius of up to 25 km.

It is known that PM particles are carriers of chemical compounds and, therefore, may negatively affect the surrounding environment and human health. Thus, research that tries to check whether such particles can also affect microbiological diversity becomes interesting.

However, some issues need the author's attention.

Line 63-65: Authors claim that the „Fungal and bacterial diversity in urban areas is lower than in rural zones [20], although the particulate matter (PM) apparently offers a protection and dispersal media for microorganisms [21,22].”

Lower fungal and bacterial diversity in urban areas is caused by fewer water bodies and exposed soils, which constitute natural reservoirs of microorganisms. Concrete and buildings do not support biodiversity. Please also explain how PM offers protection to microorganisms.

Line 102-107: At the end of the Introduction section, the purpose of the research should be stated, not a summary of the results obtained.

 Line 123: Why were only 61 samples of vegetal detritus collected compared to 70 of air and soil if three types of samples were taken from each point?

Line 127: Why was the airflow 300 l/min?

Line 129: At what height was the sampler set? Due to the different settling rates of particles depending on their size, the height from which air samples are taken for analysis will affect the size of the collected particles/microorganisms. Line 129: „After collection, aerosols were recovered by filtration through…” It would be better to write: After collection, the suspension of microorganisms and particles was filtered… Line 130: How was the filter stored after the filtration process? Line 131: Did the collected soil samples always have the same mass or volume? Line 133: How was vegetal detritus collected? How was their uniformity ensured at each measurement point?  General comments to the Methodology section: 1.       The research aimed to correlate fungal communities' diversity with PM presence.Why weren't the concentration and distribution of PM in the air measured at all measurement points? 2.       Was the terrain in front of each measurement point the same? The shape of the terrain (hills, vegetation) may influence the deposition process. Line 140: Please explain the ERA-540 abbreviation. Line 251: The legends in Figures 3a,c,d,e should be of better resolution Line 439-441: Please describe in more detail what the PM10 decontamination plan is and why, without PM concentration measurements, it was concluded that PM10 is not influenced by the mine, at least in terms of fungal distribution.

Round 2

Reviewer 2 Report

Comments and Suggestions for Authors

Thank you for your replies. I recommend the manuscript for publication.